# Cultivated Grassland Types Differently Affected Carbon Flux Downstream of the Yellow River

Yibo Wang [†], Xudong Qu [†], Meixuan Li, Juan Sun and Zhenchao Zhang *

Key Laboratory of National Forestry and Grassland Administration on Grassland Resources and Ecology in the Yellow River Delta, College of Grassland Science, Qingdao Agricultural University, Qingdao 266109, China; yibowang@stu.qau.edu.cn (Y.W.); quxudong@stu.qau.edu.cn (X.Q.); lmx99@stu.qau.edu.cn (M.L.); sunjuan@qau.edu.cn (J.S.)

* Correspondence: zhenchaozhang@qau.edu.cn; Tel.: +86-130-6121-8800
† These authors contributed equally to this work.

**Abstract:** Cultivated grasslands are an important part of grassland ecosystems and have been proven to be major carbon sinks, then playing an important role in the global carbon balance. The effect of cultivated grassland type (*Medicago sativa*, *Triticum aestivum*, *Secale cereale*, and *Vicia villosa* grasslands) on carbon flux (including net ecosystem $CO_2$ exchange (NEE), ecosystem respiration (ER), and gross ecosystem productivity (GEP)) downstream of the Yellow River was studied via the static chamber technique and a portable photosynthetic system. Bare land was used as a control. The results showed that the four cultivated grassland types were mainly carbon sinks, and bare land was a carbon source. The cultivated grassland types significantly affected carbon flux. The average NEE and GEP of the grassland types were in the following order from high to low: *Medicago sativa*, *Secale cereale*, *Triticum aestivum*, and *Vicia villosa* grassland. Stepwise regression analysis showed that among all measured environmental factors, soil pH, soil bulk density (BD), soil organic carbon (SOC), and soil microbial carbon (MBC) were the main factors affecting $CO_2$ flux. The combined influence of soil BD, SOC, and pH accounted for 77.6% of the variations in NEE, while soil BD, SOC, and MBC collectively explained 79.8% of changes in ER and 72.9% of the changes in GEP. This finding indicates that *Medicago sativa* grassland is a cultivated grassland with a high carbon sink level. The changes in carbon flux were dominated by the effects of soil physicochemical properties.

**Keywords:** grassland type; net ecosystem $CO_2$ exchange; gross ecosystem productivity; ecosystem respiration; soil physicochemical properties

## 1. Introduction

An increase in atmospheric $CO_2$ concentration leads to global warming. The carbon cycle of global terrestrial ecosystems and the main factors of the carbon cycle have become the core factors affecting current global climate change [1]. Global warming has an effect on temperature- and water-related biological processes, which in turn affect the terrestrial ecosystem carbon cycle [2]. The terrestrial ecosystem is an important part of the global carbon cycle. Plants fix $CO_2$ through photosynthesis and synthesize organic matter stored in plant roots or soil. The respiration of soil microorganisms, soil animals, and plant roots releases $CO_2$ into the atmosphere [3]. Ecosystem carbon flux is composed of net ecosystem $CO_2$ exchange (NEE), ecosystem respiration (ER), and gross ecosystem photosynthesis (GEP), indicating the absorption, transport, transformation, and synthesis of carbon among the atmosphere, plants, and soil. which directly affect the global carbon balance [4]. NEE studies in ecosystems have confirmed the balance between the number of carbon sinks on land and the number of carbon sources of $CO_2$ in the atmosphere [5]. ER is the amount of $CO_2$ released from the ecosystem into the atmosphere, including plant aboveground respiration and soil respiration [6]. Effectively controlling $CO_2$ emissions and increasing carbon sequestration are effective strategies to mitigate the trend of climate warming [7].

Grassland can store a large amount of carbon, accounting for about 20% of the total land area [8]. A change in the carbon cycle will significantly affect the carbon cycle between the terrestrial ecosystem and the atmosphere [9]. Over the past 100 years, grasslands around the world have absorbed 113.9 Pg equivalent of $CO_2$ [10]. The grassland ecosystem shows the characteristics of a carbon sink in the global carbon cycle [11]. Cultivated grassland is the main component of grassland ecosystems [12]. Cultivated grassland can not only solve the problem of forage shortage in winter but also improve the quality of forage for livestock [13]. At the same time, artificial planting and cultivation of grassland can also improve soil fertility, prevent soil erosion, and have good ecological benefits while obtaining economic benefits [14]. Cultivated grassland accounts for 28.6% of the total grassland area in the United States, 69.1% in New Zealand, and 58% in Australia [15]. Cultivated grassland has many advantages such as accumulating organic carbon, improving water conductivity and water retention, intercepting rainfall, and improving water use efficiency. Establishing cultivated grassland is an effective method to restore vegetation and improve soil fertility by accumulating soil organic carbon (SOC) content [16]. Soussana et al. (2007) studied nine European cultivated grasslands with different management measures and showed that all the cultivated grasslands were carbon sinks [17]. Tang et al. (2014) studied how different management patterns could enhance the accumulation of biomass carbon in cultivated grasslands in the mountain region of southern Ningxia, China [18]. Wang Bin monitored degraded alpine meadows and artificial grasslands in the source area of three rivers for 3 years. Both artificial grasslands were carbon sinks, and planting grassland significantly improved the leaf area index and biomass, which was conducive to increasing the carbon sink capacity of the ecosystem [19]. Cultivated grasslands have been proven to be major carbon sinks [20]. For a long time, the study of cultivated grasslands has mainly considered genetic breeding and forage performance [21,22], but there are few reports on cultivated grassland carbon flux. Therefore, understanding the response of carbon flux to different types of cultivated grassland is of great significance for the carbon sink assessment of cultivated grassland in China.

Carbon flux is affected by environmental and biological factors such as temperature, precipitation, radiation, canopy coverage, and nutrient availability, and it has great spatial variability [23]. Therefore, the temporal variability of carbon fluxes under different grassland types is controlled by different factors. For example, in the eastern and central parts of the Tibetan Plateau, temperature is the main factor affecting carbon absorption [24]. The NEE in the Horqin Sandy Land is affected by precipitation [25]. On the central Qinghai–Tibetan Plateau, warming induces an increase in net C uptake in natural alpine meadows and a decrease in cultivated grassland and alpine steppe [3]. In temperate grasslands, in addition to soil substrates and moisture, soil particle size fraction and soil bulk density (BD) also significantly affect $CO_2$ emissions [26]. Therefore, it is more and more important to explore the main factors affecting changes in carbon flux.

In this study, we investigated the effect of cultivated grassland type (*Medicago sativa*, *Triticum aestivum*, *Secale cereale*, and *Vicia villosa* grasslands) on ecosystem carbon flux. These four cultivated grassland types are relatively common downstream of the Yellow River. The main purposes of this study were to (1) determine the differences in NEE, ER, and GEP between cultivated grassland types, (2) determine which type of cultivated grassland might better improve carbon sinks, and (3) identify the main environmental factors affecting carbon flux variability. We hypothesized that different cultivated grasslands had different effects on NEE, ER, and GEP downstream of the Yellow River, mainly because of the different soil physicochemical properties.

## 2. Materials and Methods

### 2.1. Study Site

This experiment was established at the Modern Agricultural High-tech Demonstration Park of Qingdao Agricultural University in Jiaozhou City, Qingdao, Shandong Province (36°26′22″ N, 120°04′43″ E) (Figure 1). The average annual temperature in this region

is 12.3 °C, which is a warm temperate monsoon climate and has the characteristics of a maritime climate. It is warm in winter and cool in summer, with rain and heat occurring during the same period and four distinct seasons. The average annual air pressure is 1015.6 MPa, and the average annual precipitation is 724.8 mm. The annual frost-free period is 204.5 days, and the annual sunshine duration is 2708 h. The temperature and precipitation during the test are shown in Figure 2. The soil type is sand ginger black soil. The soil had the following characteristics: field water capacity, 19.17%; pH, 6.4; organic matter content, 16.46 g/kg; total nitrogen content, 0.83 g/kg; total phosphorus, 0.63 g/kg; and available phosphorus, 30.90 mg/kg.

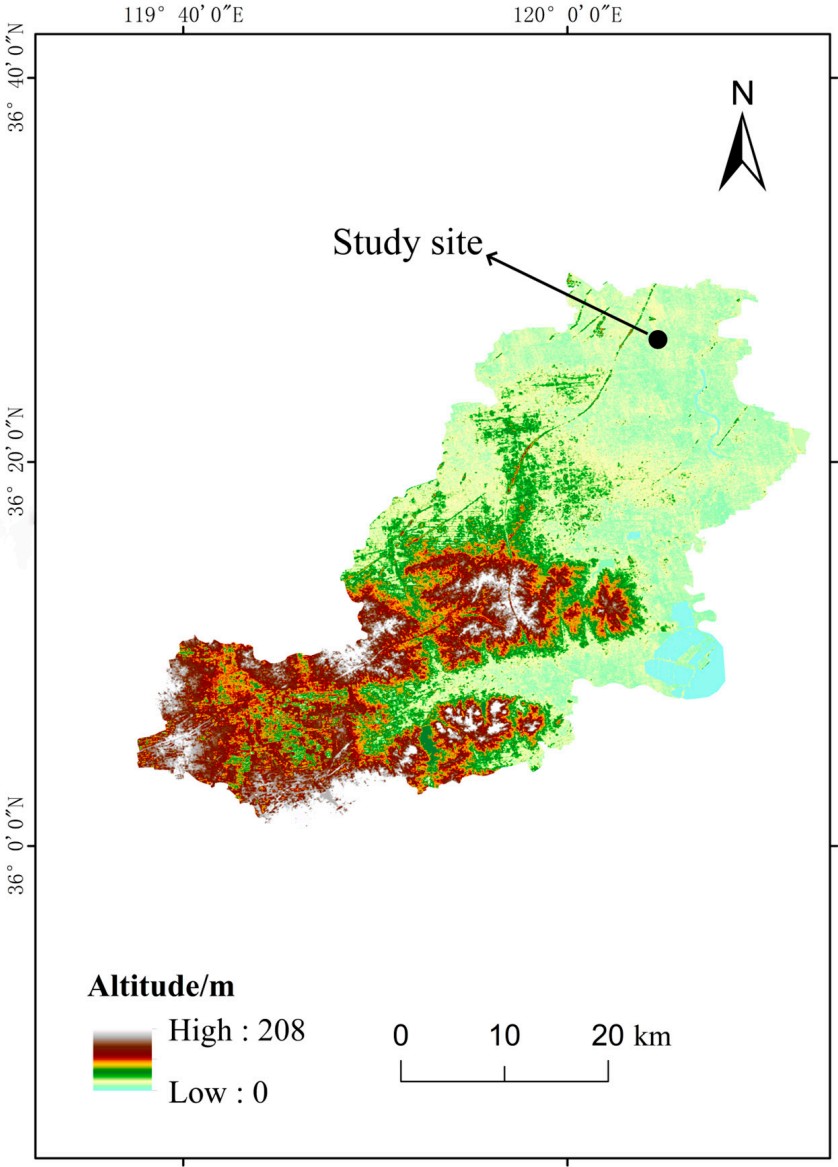

**Figure 1.** Location of the study area.

## 2.2. Experiment Design

Four experimental groups and one control group were set up in the experiment. The experimental groups were *Medicago sativa* grassland, *Triticum aestivum* grassland, *Secale cereale* grassland, *Vicia villosa* grassland, and a control group of bare land without any treatment (Figure 3). *Medicago sativa* grassland, *Triticum aestivum* grassland, *Secale cereale* grassland, and *Vicia villosa* grassland were planted in September 2022 and harvested in June 2023. A randomized block design was used in the experiment. The area of the experimental

plot was 3 × 5 m, and the planting method of drilling was used. Each plot had 4 replicates. The sowing rates of *Medicago sativa*, *Triticum aestivum*, *Secale cereale*, and *Vicia villosa* were 22.5 kg hm$^{-2}$, 150 kg hm$^{-2}$, 150 kg hm$^{-2}$, and 45.0 kg hm$^{-2}$, respectively. The row spacing was 20 cm, 20 cm, 20 cm, and 40 cm, respectively. The sowing depth was 3 cm. Weeds were removed regularly, and the plants were watered after sowing before the overwintering and regreening stages. Starting from September 2022, 50 × 50 cm quadrats representing plant conditions were selected in each plot, and a square iron frame (assimilation box base) was fixed in the soil at a depth of 5 cm. NEE, ER, and GEP were measured in all plots during the whole growth period of the plant.

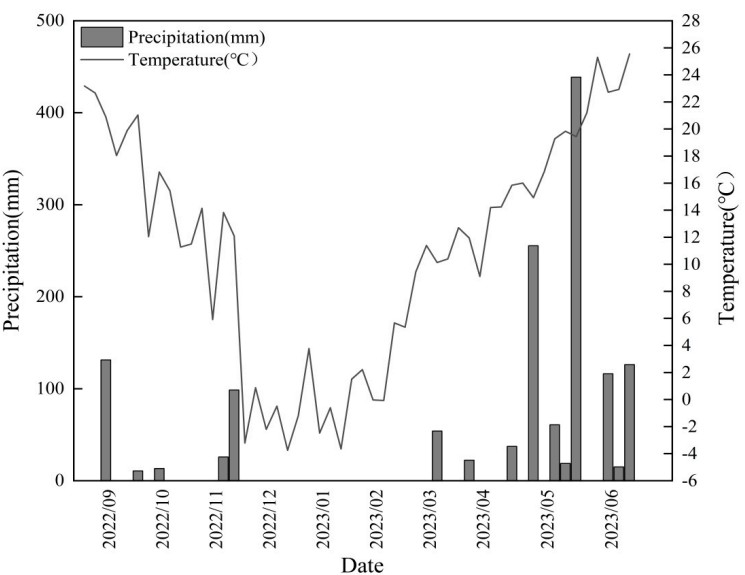

**Figure 2.** Temperature and precipitation in the study area.

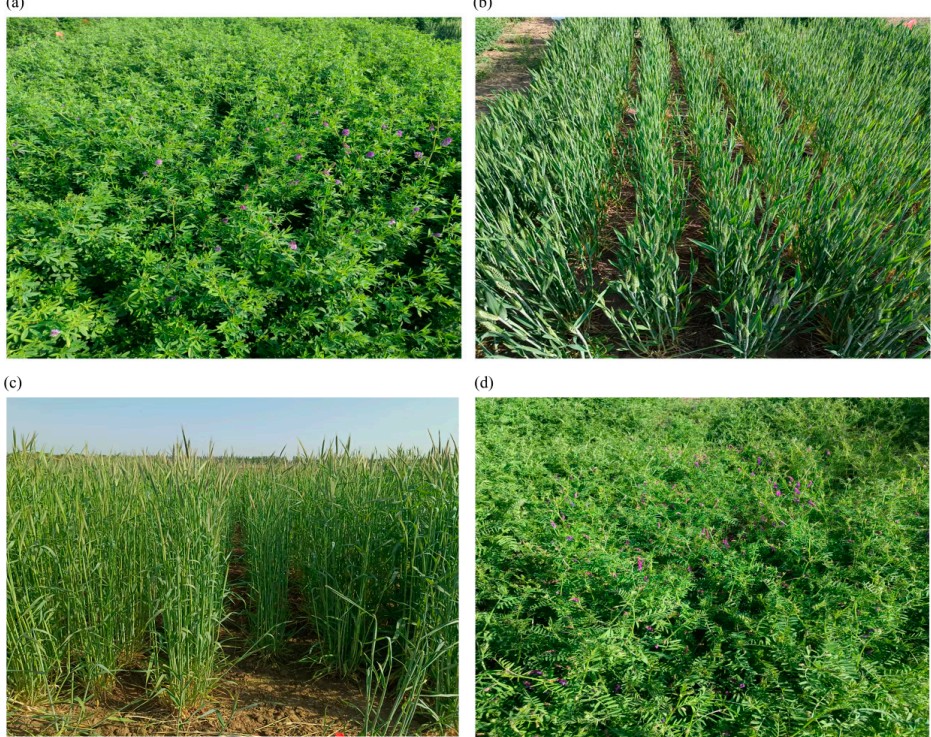

**Figure 3.** Photographs of sampling plots with different cultivated grasslands: (**a**) *Medicago sativa* grassland; (**b**) *Triticum aestivum* grassland; (**c**) *Secale cereale* grassland; and (**d**) *Vicia villosa* grassland.

### 2.3. Measurement of CO₂ Flux

During the growing seasons in 2022 and 2023, NEE was measured using an LI-6800 photosynthesis system (LI-6800XP, Li-cor, Lincoln, NE, USA). A photosynthetic assimilation box with a specification of $50 \times 50 \times 100$ cm was made, and four fans were installed to mix the gas in the box. The LI-6800 host was connected to the assimilation box during the measurement. The program was set to record data every 10 s, and 12 data points were recorded continuously. Clear and cloudless weather was selected for measurement. The measurement time was 8:00–12:00 a.m. After NEE was measured, the side opening of the assimilation box was placed for about 1–2 min to fully mix the gas in the assimilation box. After that, the assimilation box was placed on the base of the assimilation box, covered with a shading cloth, and ER was measured [27]. NEE and ER were calculated using the following equation:

$$F_C = \frac{10VP_0\left(1 - \frac{W}{1000}\right)}{RS(T_0 + 273.15)} \frac{\partial C'}{\partial t} \tag{1}$$

where $F_C$ is the CO₂ flux rate (μmol·m$^{-2}$·s$^{-1}$); $V$ is the volume of the assimilation box (cm³); $P_0$ is the average atmospheric pressure in the chamber (kPa); $W$ is the water vapor concentration in the chamber (mmol·mol$^{-1}$); $R$ is the ideal gas constant (8.314 J·mol$^{-1}$·K$^{-1}$); $S$ is the bottom area of the box (cm²); $T_0$ is the average temperature in the box; and $\partial C'/\partial t$ is the rate at which the CO₂ concentration in the box changes with time.

GEP was calculated using the following equation:

$$\text{GEP} = \text{NEE} - \text{ER} \tag{2}$$

### 2.4. Soil Physicochemical Property Measurement

The soil pH was determined using a pHS-3G digital pH meter (from Shanghai LeiMag Instrument Factory, Shanghai, China). The soil BD was determined using the ring tool method [28]. The SOC content was determined using the potassium dichromate external heating method [28]. To determine the soil total nitrogen and carbon content, the soil was finely ground using a ball mill and analyzed using an elemental analyzer (Elementar, Vario EL cube, Germany) [29]. The soil microbial biomass carbon was extracted and determined using chloroform fumigation [30]. Soil nitrate nitrogen and ammonium nitrogen were extracted with 2 mol·L$^{-1}$ potassium chloride and analyzed using a continuous flow analyzer (SEAL, AA3, Germany) [31].

### 2.5. Statistical Analysis

Microsoft Excel 2010 was used to organize the original data, and SPSS 22.0 software was used to perform a one-way analysis of variance on the original data. The Duncan method was used for multiple comparisons. All data were expressed as mean ± standard error.

Regression analysis was used to compute the relationship between CO₂ flux and soil physicochemical properties.

## 3. Results

### 3.1. Treatment Effects on NEE, ER and GEP

There were significant differences in NEE among the treatments. The NEE of *Medicago sativa* grassland was significantly higher than that of other grassland types, reaching the maximum value of 18.42 μmol·m$^{-2}$·s$^{-1}$ in May. The NEE of *Triticum aestivum*, *Secale cereale*, and *Vicia villosa* grasslands reached the maximum value in April, with values of 12.73, 13.27, and 11.40 μmol·m$^{-2}$·s$^{-1}$, respectively (Figure 4a). The average NEE fluxes of *Medicago sativa*, *Triticum aestivum*, *Secale cereale*, and *Vicia villosa* grasslands were significantly higher than that of the bare land by 6.41, 3.88, 4.27, and 2.79 μmol·m$^{-2}$·s$^{-1}$, respectively ($p < 0.05$; Figure 5a). The analysis of the NEE throughout the growing period showed that the

grassland types were in the following order: *Medicago sativa* grassland > *Secale cereale* grassland > *Triticum aestivum* grassland > *Vicia villosa* grassland > bare land (Figure 5a).

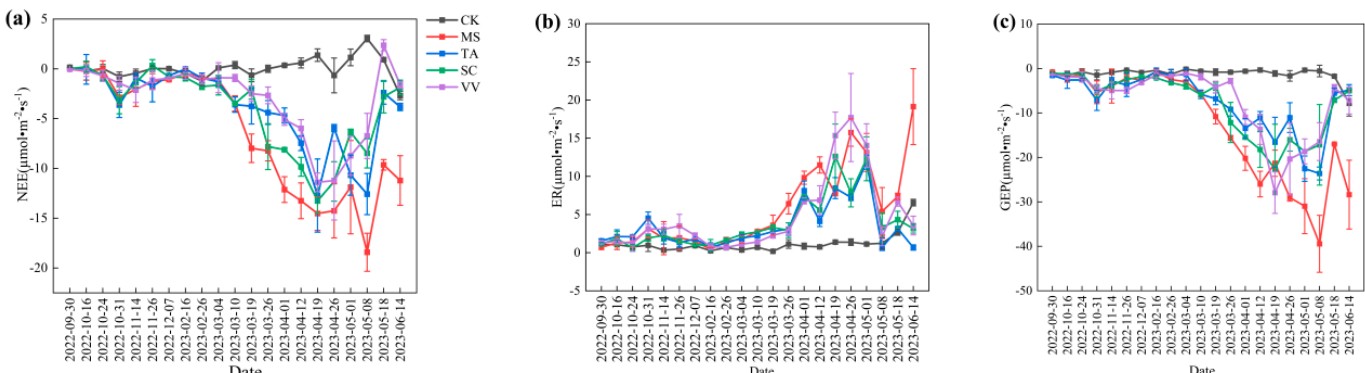

**Figure 4.** Dynamics of net ecosystem $CO_2$ exchange (NEE) (**a**), ecosystem respiration (ER) (**b**), and gross ecosystem productivity (GEP) (**c**) during the 2022–2023 growing period. CK: bare land; MS: *Medicago sativa* grassland; TA: *Triticum aestivum* grassland; SC: *Secale cereale* grassland; and VV: *Vicia villosa* grassland. Values are the mean ± standard error. The unit of $CO_2$ flux is $\mu mol \cdot m^{-2} \cdot s^{-1}$. The same applies below.

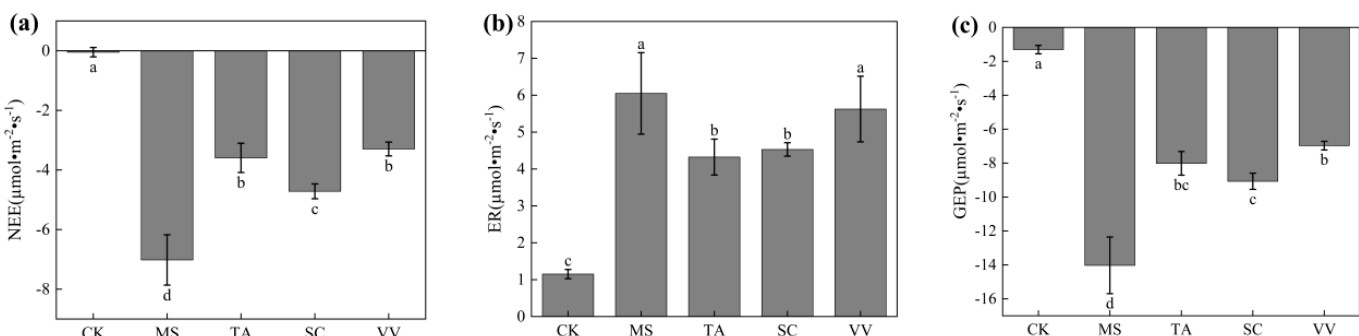

**Figure 5.** Mean values of NEE (**a**), ER (**b**), and GEP (**c**) in the growing season of bare land and cultivated grasslands. Error bars show the standard error.

The overall change trend of ER in bare land was not obvious, and it reached the maximum value of 6.56 $\mu mol \cdot m^{-2} \cdot s^{-1}$ in June. The ER of the four cultivated grasslands gradually increased after March 19 in the growing season, and the ER of *Medicago sativa* grassland reached its maximum value in June (19.13 $\mu mol \cdot m^{-2} \cdot s^{-1}$). The ER of *Triticum aestivum* grassland reached its maximum value in May (11.85 $\mu mol \cdot m^{-2} \cdot s^{-1}$). The ER of *Secale cereale* grassland reached its maximum value in April (12.62 $\mu mol \cdot m^{-2} \cdot s^{-1}$). The ER of *Vicia villosa* grassland peaked at 17.70 $\mu mol \cdot m^{-2} \cdot s^{-1}$ in April (Figure 4b). The ER of *Medicago sativa* and *Vicia villosa* grasslands was significantly higher than that of other treatments ($p < 0.05$; Figure 5b). The analysis of the ER throughout the growing period showed that the grassland types were in the following order: *Medicago sativa* grassland > *Vicia villosa* grassland > *Secale cereale* grassland > *Triticum aestivum* grassland > bare land (Figure 5b).

There were significant differences in GEP among grassland types. The GEP of *Medicago sativa* grassland reached its maximum value in June (28.37 $\mu mol \cdot m^{-2} \cdot s^{-1}$). The GEP of *Triticum aestivum* grassland reached its maximum value on May 1 (22.53 $\mu mol \cdot m^{-2} \cdot s^{-1}$). The GEP of *Secale cereale* grassland reached its maximum value in April (22.57 $\mu mol \cdot m^{-2} \cdot s^{-1}$). The GEP of *Vicia villosa* grassland reached its maximum value in April (27.93 $\mu mol \cdot m^{-2} \cdot s^{-1}$). The GEP of *Medicago sativa* grassland was significantly higher than that of other treatments ($p < 0.05$; Figure 4c). The average GEP contents of *Medicago sativa*, *Triticum aestivum*, *Secale cereale*, and *Vicia villosa* grasslands were higher than that of the bare land by 11.52, 5.80,

7.04, and 5.18, $\mu mol \cdot m^{-2} \cdot s^{-1}$, respectively ($p < 0.05$; Figure 5c). The analysis of the GEP throughout the growing period showed that the grassland types were in the following order: *Medicago sativa* grassland > *Secale cereale* grassland > *Triticum aestivum* grassland > *Vicia villosa* grassland > bare land (Figure 5c).

### 3.2. Treatment Effects on Soil Physicochemical Properties

Different grasslands had different effects on soil properties at a soil depth of 0–10 cm (Table 1). There were no significant differences in soil pH among the treatments. The soil BD of *Medicago sativa* grassland was significantly lower than that of the other treatments ($p < 0.05$). The SOC content of bare land was significantly lower than that of the other treatments ($p < 0.05$). *Secale cereale* grassland had the highest organic carbon content. There were no significant differences in soil TN content among the different treatments. There were no significant differences in soil $NH_4^+$ content among treatments ($p > 0.05$). The soil $NO_3^-$-N content of *Triticum aestivum*, *Secale cereale*, and *Vicia villosa* grasslands was significantly lower than that of bare land and *Medicago sativa* grassland ($p < 0.05$). The soil MBC content of bare land was significantly lower than that of the other treatments ($p < 0.05$). The soil MBC content of *Medicago sativa* grassland was significantly higher than that of *Secale cereale* and *Vicia villosa* grasslands ($p < 0.05$).

**Table 1.** Main soil properties under different treatments.

| Treatments | pH | BD ($g \cdot cm^{-3}$) | SOC ($g \cdot kg^{-1}$) | TN ($g \cdot kg^{-1}$) | $NH_4^+$ -N ($mg \cdot kg^{-1}$) | $NO_3^-$ N ($mg \cdot kg^{-1}$) | MBC ($mg \cdot kg^{-1}$) |
|---|---|---|---|---|---|---|---|
| CK | 6.33 ± 0.54 a | 1.68 ± 0.05 a | 5.52 ± 0.79 b | 1.24 ± 0.04 a | 5.45 ± 0.95 a | 7.85 ± 3.16 a | 237.68 ± 27.80 c |
| MS | 5.87 ± 0.40 a | 1.53 ± 0.11 b | 10.26 ± 0.54 a | 1.23 ± 0.12 a | 6.37 ± 1.13 a | 9.80 ± 1.95 a | 1030.02 ± 44.29 a |
| TA | 6.17 ± 0.50 a | 1.61 ± 0.08 ab | 11.64 ± 1.67 a | 1.28 ± 0.06 a | 6.61 ± 0.44 a | 1.43 ± 0.26 b | 873.50 ± 166.35 ab |
| SC | 6.14 ± 0.36 a | 1.61 ± 0.02 ab | 11.83 ± 0.03 a | 1.26 ± 0.03 a | 6.32 ± 1.05 a | 1.24 ± 0.09 b | 691.76 ± 51.21 b |
| VV | 6.24 ± 0.45 a | 1.63 ± 0.14 ab | 11.12 ± 0.60 a | 1.25 ± 0.02 a | 7.72 ± 2.01 a | 4.05 ± 2.50 b | 770.31 ± 175.48 b |

Data are presented as the mean ± standard error. Different letters indicate a significant difference among treatments ($p < 0.05$). BD: bulk density; SOC: soil organic carbon; TN: total nitrogen; $NH_4^+$-N: ammonium nitrogen; $NO_3^-$-N: nitrate nitrogen; and MBC: microbial biomass carbon. The same applies below.

### 3.3. Factors Affecting $CO_2$ Flux

Linear regression analysis showed that NEE had a significantly positive correlation with SOC ($p < 0.01$) and a negative correlation with soil pH ($p < 0.01$) and soil BD ($p < 0.05$; Figure 6). SOC explained 35.1% of the variation in NEE (Figure 6c). Soil pH explained 21.0% of the variation in NEE (Figure 6b). Soil BD explained 24.7% of the variation in NEE (Figure 6a). ER had a significant positive correlation with SOC ($p < 0.01$; Figure 6d) and MBC ($p < 0.01$; Figure 6e). SOC explained 48.2% of the variation in ER (Figure 6d). MBC explained 60.1% of the variation in ER (Figure 6e). GEP had a negative correlation with soil BD ($p < 0.05$; Figure 6f). GEP had a significant positive correlation with SOC ($p < 0.01$; Figure 6g) and MBC ($p < 0.01$; Figure 6h). Soil BD explained 28.8% of the variation in GEP (Figure 6f). SOC explained 41.9% of the variation in GEP (Figure 6f). MBC explained 75.4% of the variation in GEP (Figure 6h).

The NEE, ER, and GEP rates and major environmental factors were analyzed by stepwise regression. Soil BD, SOC, and pH explained 77.6% of the variation in NEE, while BD, SOC, and MBC explained 79.8% and 72.9% of the variation in NEE and GEP (Table 2).

**Table 2.** Stepwise regression equations of NEE, ER, GEP, and their factors.

| Dependent Variables | Step Regression Equation | $R^2$ | $p$ |
|---|---|---|---|
| NEE | $y = -11.653x_1 + 0.409x_2 - 1.847x_3 + 29.454$ | 0.776 | <0.01 |
| ER | $y = -8.160x_1 + 0.316x_2 + 0.003x_4 + 12.070$ | 0.798 | <0.05 |
| GEP | $y = -19.954x_1 + 0.663x_2 + 0.005x_4 + 29.025$ | 0.729 | <0.05 |

xx$_1$: BD; x$_2$: SOC; x$_3$: pH; x$_4$: MBC.

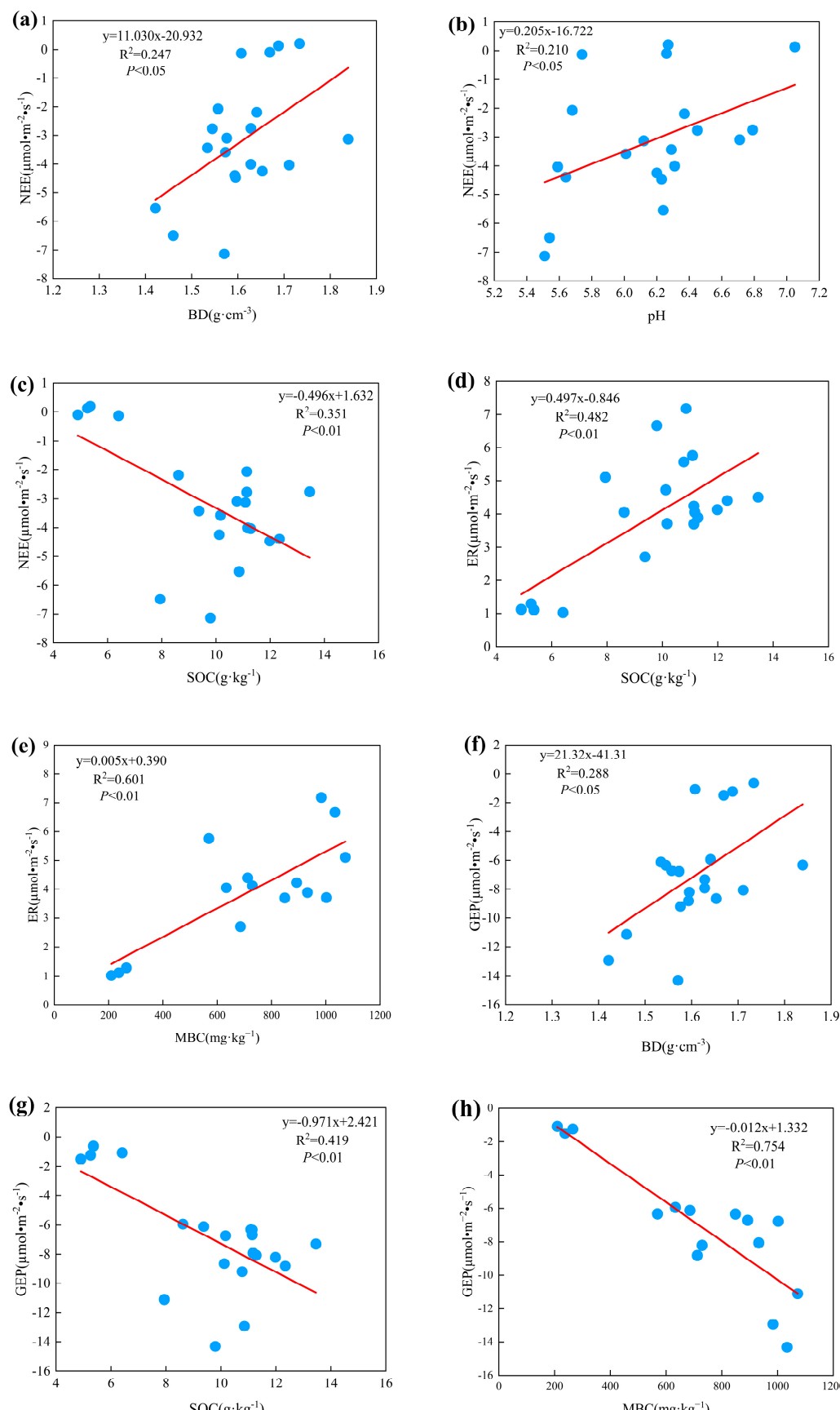

**Figure 6.** Correlation between NEE and BD (**a**), pH (**b**) and SOC (**c**) contents. Correlation between ER and SOC (**d**) and MBC (**e**). Correlation between GEP and BD (**f**), SOC (**g**) and MBC (**h**).

## 4. Discussion

### 4.1. Variation in the Characteristics of Carbon Flux in Cultivated Grasslands

NEE reflects the carbon source/sink strength of the ecosystem. When NEE is positive, the ecosystem emits $CO_2$ into the atmosphere, and when it is negative, the ecosystem fixes $CO_2$ from the atmosphere [24]. In this study, the four cultivated grasslands were all carbon sinks. Our research demonstrated that as seasons changed, the NEE and GEP of all cultivated grasslands initially increased and then decreased, and the maximum values appeared in April and May. When the temperature is gradually rising, active microorganisms decompose more litter and increase nutrient accumulation in the soil, enhancing photosynthesis. These positive effects result in a gradual increase in the total productivity of the ecosystem [32]. The soil temperature increased from April to May. ER is composed of plant autotrophic respiration and soil microbial heterotrophic respiration. Elevated soil temperature promotes microbial activity and rapidly decomposes soil organic carbon, resulting in an increase in ER [33]. In June, except for *Medicago sativa*, other plants entered the end of the growing season, soil microorganisms and root respiration weakened, and ER significantly decreased. In this study, the ER of *Medicago sativa* grassland and *Vicia villosa* grassland was significantly higher than that of *Triticum aestivum* grassland and *Secale cereal* grassland, and the grassland coverage of *Medicago sativa* grassland and *Vicia villosa* grassland was higher than that of *Triticum aestivum* and *Secale cereal* grassland. Soil surface density is conducive to creating an aerobic environment, which leads to an increase in ER [34].

### 4.2. Effects of Environmental Factors on Carbon Fluxes

Soil physicochemical properties that were shown to significantly affect NEE, ER, and GEP included soil BD, pH, and SOC. In this study, NEE, ER, and GEP were significantly positively correlated with SOC. SOC is an important part of the carbon cycle. In the process of plant growth, increases in aboveground and underground biomass promoted the accumulation of soil carbon [35]. Similarly, SOC reacts to plants. The increase in SOC promotes plant growth [36] and plants absorb more $CO_2$ through photosynthesis during growth [37], thus increasing NEE and GEP. Artificial grass planting can increase the SOC content. With an increase in root biomass, the content of root exudates also increases significantly. This process can promote the decomposition of organic carbon by soil microorganisms. Plant roots penetrate the soil agglomeration structure, reducing the protective effect of organic carbon, making more organic carbon participate in the activity and metabolism of microorganisms, intensifying the decomposition process of organic carbon, and thus releasing more $CO_2$ [38]. SOC is an important substrate for soil microbial respiration. With an increase in temperature, the higher the SOC content, the higher the soil microbial respiration rate, and the higher the soil efficiency [39]. Franzluebbers showed that soil respiration was strongly regulated by carbon substrates in soil organic matter and that there was a linear correlation between soil basic respiration and SOC content [40]. Liu showed that a high SOC content was usually related to high $CO_2$ emissions due to an enhanced soil labile C content and C turnover rate [41]. This is consistent with the results of our study.

As a negative factor for NEE (Figure 6b), the soil pH suggests that soils with a low pH are more suitable for enhancing NEE. This was also confirmed by the fact that the soil pH was lowest in *Medicago sativa* grassland, and NEE was highest in the four cultivated grasslands in this study. Soil pH affects the absorption of soil nutrients by plants. During the growth of plants, physiological and biological characteristics also affect soil pH [42]. The reduction in soil pH caused by planting *Medicago sativa* is attributed to the strong nitrogen-fixing ability of *Rhizobium* in the alfalfa rhizosphere, which leads to the secretion of $H^+$ ions and various organic acids by nitrogen-fixing bacteria and roots, thereby lowering the soil pH [43], consequently affecting soil microorganisms and plants, and indirectly affecting ecosystem C cycling [44]. The soil pH can affect the effective utilization of soil elements by plants, but the increase in soil pH limits the absorption of soil nutrients, affecting

plant growth and productivity [45] and therefore affecting NEE and GEP. In this study, the soil BD of alfalfa was significantly lower than that of the other treatments (Table 1). The results showed that planting *Medicago sativa* reduced the soil BD and enhanced soil permeability, which was more beneficial to plant growth. Alfalfa develops a taproot, more roots, and longer roots, meaning that it can play a very good role in thinning soil during the development and extension of roots [46]. In this study, soil BD had a significant effect on NEE and GEP, accounting for 24.7% and 28.8% of the variation in NEE and GEP, respectively. Excessive soil BD can inhibit plant root growth and mineral nutrient uptake, thereby reducing the plant photosynthetic rate and aboveground productivity [47] as well as NEE and GEP. *Medicago sativa* grassland had the smallest soil BD and the largest soil porosity, promoting the absorption of soil nutrients by plants [48] and increasing the aboveground biomass [49], microbial activity [50], NEE, and GEP.

Soil microbial carbon content can reflect the quantity and activity of soil microorganisms, which directly affect the activity of soil microorganisms and their roots, and then affect soil respiration. When soil microbial carbon content increases, microbial activity and quantity increase, leading to the decomposition of organic matter by microorganisms. In this process, microorganisms consume organic matter through respiration and release gases such as carbon dioxide, which leads to ER enhancement [51]. The root system of alfalfa is developed, and the root system secretes a variety of substances during the growth process. These root exudates have a significant effect on the soil environment. The root exudates are rich in organic substances. These organic substances gradually accumulate in the soil, thereby increasing the content of soil organic carbon and microbial carbon and providing sufficient carbon sources for the growth and reproduction of microorganisms [52], thus promoting the respiration of the aboveground and underground parts.

## 5. Conclusions

There were significant differences in carbon flux among the cultivated grassland types. Through the NEE measurement during the growth period of four kinds of grassland, all four kinds of grassland showed carbon sink. *Medicago sativa* grassland was beneficial in increasing the carbon sink. Soil physicochemical properties in cultivated grasslands had significant effects on the $CO_2$ flux. The combined influence of soil BD, SOC, and pH accounted for 77.6% of the variations in NEE across different cultivated grasslands, while soil BD, SOC, and MBC collectively explained 79.8% of changes in ER and 72.9% of changes in GEP. By measuring the change characteristics of carbon fluxes during the growth period of several commonly cultivated grasslands downstream of the Yellow River, our study relatively accurately estimated the $CO_2$ exchange fluxes of these cultivated grasslands, providing theoretical support for the development of emission reduction measures in this region. The cultivated grassland area and the importance of cultivated grassland carbon sink functions will continue to increase. In the case of intensified climate change in the future, the characteristics of cultivated grassland carbon flux change should be comprehensively affected by more factors. Therefore, long-term monitoring and a more comprehensive and in-depth exploration are needed to reveal its internal mechanism.

**Author Contributions:** Conceptualization, Y.W. and X.Q.; methodology, Y.W.; software, X.Q.; validation, M.L.; formal analysis, Y.W.; investigation, M.L.; resources, J.S.; data curation, X.Q.; writing—original draft preparation, Y.W.; writing—review and editing, J.S.; visualization, X.Q.; supervision, Z.Z.; project administration, J.S.; funding acquisition, J.S. All authors have read and agreed to the published version of the manuscript.

**Funding:** This research was funded by the China Agriculture Research System (CARS-34).

**Data Availability Statement:** The original contributions presented in the study are included in the article, further inquiries can be directed to the corresponding author.

**Acknowledgments:** The authors are thankful for the support of the foundation and thank the teachers and students for their help and guidance.

**Conflicts of Interest:** The authors declare that they have no known competing financial interests or personal relationships that could have appeared to influence the work reported in this paper.

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
