# Peer review of "Cultivated Grassland Types Differently Affected Carbon Flux Downstream of the Yellow River"

_agronomy, doi:10.3390/agronomy14050974_

Round 1
Reviewer 1 Report
Comments and Suggestions for Authors
The paper titled “Cultivated Grassland Types Differently Affected Carbon 2 Sequestration Downstream of the Yellow River” is based on experiment and may has potential to contribute knowledge specially role of grasslands for carbon dynamics. The paper may be considerd with following modifications.
1. Abstract is described poorly. Abstract is full of redundant. Please be precise for rational, objective, method of data collection, (i.e. number of survey, what has been collected etc.), method applied (name) and then result. Please rewrite the abstract.
2. Keyword should be unique to trace the paper by others and should not be include those words which are included already in title. Please modify.
3. Language needs to be relooked across the MS.
4. Please justify why you have interested for the comparison. Please provide the area under each grasslands. This will make your rational sound and further justify the gap in knowledge. Please also highlights the contribution of vegetation for Co2 and what is the contribution of grasslands across globe and your country.
5. The overall questions (focusing to comparison) is weakly articulated. Moreover please rewrite questions, hypothesis and objectives.
6. The details of all plots along with their location details, climate, anthropogenic pressure needs to eb flagged in supplementary material.
7. Experimental settings has not been described well. Pl clearly mentioned what was the treatments, no fo replication and design of experiments. How you ensure randomization, needs to be flagged. Please also provide details about area of plots along with management aspects including climatic details and time period for the experiment. What was the management protocol for the weeding, watering etc. Pl also justify why you have selected these four gasses and why you are comparing with barren lands. How barren lands would be a source.
8. PL justify why you have applied one way. Pl do two-way ANOVA considering replication also as factor. Pl also describe the regression. You may attempt stepwise regression. You may refer Large Scale Spatial Assessment, Modelling and Identification of Drivers of Soil Respiration.
9. Please mention how many measurements you have observed for each variable for each grasslands.
10. Please provide the details of calibration for each used instruments.
11. Pl elaborate the quality of grasses i.e. growth and survivality etc.
12. The result is presented well however I would like to see the causes of differences that should not be based simply the difference of the grasses rather other attributal causes also, if feasible.
13. Please check title of each Table and figure. Figure 6 is scatter plot with regression equation. If you at all interested for evaluation the drivers, pl use syep wise regress. It would be good for better result.
14. Though the discussion is attempted to account the result however poorly describe. Pl improve based on study on similar aspects in similar region.
15. How the study is useful for other regions needs to be included.
16. Please include limitation and future researches directions.
17. Please make precise the conclusion.
Comments on the Quality of English LanguageMay be improved
Author Response
Dear reviewer:
Thank you for your letter and for the reviewers’ comments concerning our manuscript entitled "Cultivated Grassland Types Differently Affected Carbon Sequestration Downstream of the Yellow River" (ID:agronomy-2964681). Those comments are all valuable and very helpful for revising and improving our paper, as well as the important guiding significance to our researches. We have studied comments carefully and have made correction which we hope meet with approval. The main corrections in the paper and the responds to the editor’s and the reviewer’s comments are as flowing:
Responds to the reviewer’s comments:
- Abstract is described poorly. Abstract is full of redundant. Please be precise for rational, objective, method of data collection, (i.e. number of survey, what has been collected etc.), method applied (name) and then result. Please rewrite the abstract.
Response: Thanks for your comments which are important to improve our manuscript. As the reviewer suggested, the abstract has been revised as follows:
Cultivated grassland is an important part of grassland ecosystem and has proven to be major carbon sinks. Cultivated grasslands play an important role in the global carbon balance. The effect of cultivated grassland type (Medicago sativa, Triticum aestivum, Secale cereale, and Vicia villosa grasslands) on carbon flux (including net ecosystem CO2 exchange (NEE), ecosystem respiration (ER), and gross ecosystem productivity (GEP) downstream of the yellow river was studied via static chamber technique and the portable photosynthetic system. Bare land was used as a control. The results showed that the four cultivated grassland types were mainly carbon sinks, and bare land was a carbon source. The cultivated grassland types significantly affected the carbon flux. The average NEE and GEP of the grassland types were in the following order from high to low: Medicago sativa, Secale cereale, Triticum aestivum, and Vicia villosa grassland. Stepwise regression analysis showed that among all measured environmental factors, soil pH, soil bulk density(BD), and soil organic carbon(SOC) and soil microbial carbon(MBC) were the main factors affecting CO2 flux. The combined influence of soil BD, SOC and pH accounted for 77.6% of the variations in NEE, while soil BD, SOC, and MBC collectively explained 79.8% of changes in ER and 72.9% of changes in GEP. This finding indicates that Medicago sativa grassland is a cultivated grassland with a high carbon sink level. The changes in carbon flux were dominated by the effects on soil physicochemical properties. Please see line 12-28.
- Keyword should be unique to trace the paper by others and should not be include those words which are included already in title. Please modify.
Response: Thanks for your comments which are important to improve our manuscript. As the reviewer suggested, the keywords have been revised as follows:
Keywords: Grassland type; Net ecosystem CO2 exchange; Gross ecosystem
productivity; Ecosystem respiration; Soil physicochemical properties
Please see line 29-30.
- Language needs to be relooked across the MS.
Response: Thanks for your comments. We have modified it in the revised manuscript.
- Please justify why you have interested for the comparison. Please provide the area under each grasslands. This will make your rational sound and further justify the gap in knowledge. Please also highlights the contribution of vegetation for CO2and what is the contribution of grasslands across globe and your country.
Response: Thanks for your comment. In this study, we investigated the effect of cultivated grassland type (Medicago sativa, Triticum aestivum, Secale cereale, and Vicia villosa grasslands) on ecosystem carbon flux. These four cultivated grassland types are relatively common downstream of the Yellow River. The main purposes of this study were to (1) determine the differences in NEE, ER, and GEP between cultivated grassland types, (2) determine which type of cultivated grassland might better improve carbon sinks, and (3) identify the main environmental factors affecting carbon flux variability. Carbon flux under different grassland types have shown significantly different daily and seasonal variations. It’s critical to investigate carbon flux rates across different grassland types in order to better understand the geographic and temporal patterns of carbon flux rates, as well as the significance of local carbon sources and sinks.
The contribution of vegetation to carbon dioxide is mainly reflected in its absorption and storage functions. Through photosynthesis, plants are able to absorb carbon dioxide from the atmosphere, convert it into organic matter and release oxygen. This process not only provides the plants with the energy and nutrients they need to grow, but also helps to reduce the concentration of carbon dioxide in the atmosphere, which is crucial to achieving the goal of carbon neutrality. The contribution of vegetation to carbon dioxide is mainly reflected in its absorption and storage functions. Through photosynthesis, plants are able to absorb carbon dioxide from the atmosphere, convert it into organic matter and release oxygen. This process not only provides the plants with the energy and nutrients they need to grow, but also helps to reduce the concentration of carbon dioxide in the atmosphere, which is crucial to achieving the goal of carbon neutrality. There are differences in CO2 emissions among ecosystems and vegetation kinds. For example, forest ecosystems are higher than grassland ecosystems, Stipa baicalensis steppe is higher than Leymus chinensis steppe, and scrub > interclump meadow > bare ground is shown in alpine scrub.
Grassland ecosystems, as a pillar of terrestrial ecosystems, occupy about 20% of the total land area. In the global terrestrial ecosystem, grassland ecosystem is second only to forest ecosystem in carbon storage. For grassland ecosystem, carbon is mainly stored in grassland plants, litter and soil humus. Part of the carbon fixed by grassland vegetation is used for its own growth and reproduction, and the other part is eaten by herbivores, eaten by animals, and then returned to the soil in the form of feces. The aboveground part of the plant, which is not eaten by animals, transfers carbon to the soil by forming litter, while the underground part of the plant transfers carbon to the soil by secreting and forming plant roots. Global grassland ecosystem carbon storage is about 308 Pg C.
China 's grassland area accounts for 40 % of China 's land area and 13 % of the world 's grassland area. It is about 400 million hm2, which is 2.5 times the forest area and 3.2 times the cultivated land area. It is the largest carrier for absorbing CO2 and the largest carbon pool in China. The carbon sink potential of grassland ecosystem is great. It can not only be reflected by the biomass of vegetation such as stems, branches and leaves, but also be fixed by soil organic matter and litter. It is estimated that the annual carbon sequestration of grassland in China has reached 1 ~ 2t / hm2, which can absorb about 40 % of the annual carbon emissions, and the total carbon sequestration is about 600 million tons. In the temperate steppe of Inner Mongolia, the carbon storage of meadow steppe was the highest, and that of desert steppe was the lowest. The carbon storage of the alpine meadow on the Qinghai-Tibet Plateau is about 20 Pg.
- 5. The overall questions (focusing to comparison) is weakly articulated. Moreover please rewrite questions, hypothesis and objectives.
Response: Thanks for your comment. As the reviewer suggested, questions, hypothesis and objectives have been revised as follows:
The main purposes of this study were to (1) determine the differences in NEE, ER, and GEP between cultivated grassland types, (2) determine which type of cultivated grassland might better improve carbon sinks, and (3) identify the main environmental factors affecting carbon flux variability. We hypothesized that different cultivated grasslands had different effects on NEE,ER and GEP downstream of the yellow river, mainly because of the different soil physicochemical properties. Please see line 94-99.
- 6. The details of all plots along with their location details, climate, anthropogenic pressure needs to eb flagged in supplementary material.
Response: Thank you, the comments are very important to improve our manuscript.
Supplementary material: All plots were established at the Modern Agricultural High-tech Demonstration Park of Qingdao Agricultural University in Jiaozhou City, Qingdao, Shandong Province (36°26’22”N, 120°04’43”E). Temperature and precipitation are shown in the following figure:
Figure 1. Temporal patterns of daily mean air temperature and daily total precipitation in the study area.
- 7. Experimental settings has not been described well. Pl clearly mentioned what was the treatments, no fo replication and design of experiments. How you ensure randomization, needs to be flagged. Please also provide details about area of plots along with management aspects including climatic details and time period for the experiment. What was the management protocol for the weeding, watering etc. Pl also justify why you have selected these four gasses and why you are comparing with barren lands. How barren lands would be a source.
Response: Thanks for your comments. As suggested, we have supplemented the experimental design section by marking the manuscript in red. The experimental design has been revised as follows:
Four experimental groups and one control group were set up in the experiment. The experimental groups were : Medicago sativa grassland, Triticum aestivum grassland, Secale cereale grassland, Vicia villosa grassland, control group : bare land without any treatment. Medicago sativa grassland, Triticum aestivum grassland, Secale cereale grassland and Vicia villosa grassland were planted in September 2022 and harvested in June 2023. A randomized block design was used in the experiment. The area of the experimental plot was 3 * 5 m, and the planting method of drilling was used. Each plot had 4 replicates. The sowing rates of alfalfa, wheat, Dongmu 70 rye and hairy vetch were 22.5 kg · hm-2, 150 kg · hm-2, 150 kg · hm-2, 45.0 kg · hm-2, respectively. The row spacing was 20 cm, 20 cm, 20 cm and 40 cm, respectively. The sowing depth was 3 cm. Weeds were cleaned regularly, and watered after sowing, before overwintering and regreening stage. Starting from September 2022, 50 * 50 cm quadrats representing plant conditions were selected in each plot, and a square iron frame ( assimilation box base ) was fixed in the soil at a depth of 5 cm. NEE, ER and GEP were measured in all plots during the whole growth period of the plant. Please see line 118-132.
The main reasons for choosing these four grasslands are as follows.
- These four kinds of cultivated grasslands are common downstreamof the Yellow River.
- Medicago sativais rich in protein, minerals, vitamins and other nutrients, has a strong adaptability, can grow in a variety of climate and soil conditions, widely planted in the Yellow River basin. Secale cerealehas the characteristics of cold resistance, drought resistance, salt and alkali resistance, disease resistance, early greening, vigorous growth, strong tillering ability and regeneration ability, good palatability, and high grass yield, good yield performance and high nutritional value. Vicia villosa is the main green manure crop downstream of the Yellow River. It has the characteristics of wide adaptability, high biological yield and strong nitrogen fixation ability, which can increase the population of soil microorganisms and provide sufficient carbon source for soil microorganisms. The downstream of Yellow River has become the main wheat producing area because of its superior geographical environment and climatic conditions.
- These four plants all contribute to improving grassland productivity, but there are few studies on their carbon sequestration capacity and contribution rate to greenhouse gas emissions. By monitoring the carbon flux of these four plants during their growth period, we want to determine the characteristics of their carbon sinks and carbon emissions, so as to improve the measured data and lay a theoretical foundation for the carbon pool management of cultivated grasslandsdownstream ofthe lower Yellow River.
The purpose of setting bare land as controls is to measure the differences in carbon sinks and emissions between planted and unplanted plants under the same soil conditions. Since there is no vegetation cover, bare land cannot photosynthesize as vegetation does, absorbing carbon dioxide from the atmosphere and converting it into organic matter. On the contrary, bare land may be due to the decomposition of soil microorganisms, weathering and erosion, etc., resulting in the release of organic carbon in the soil into the atmosphere, thereby increasing the atmospheric carbon dioxide concentration.
The plot of the experiment is shown below:
CK: Bare land; MS: Medicago sativa grassland; TA: Triticum aestivum grassland; SC: Secale cereale grassland; VV:Vicia villosa grassland.
- 8. PL justify why you have applied one way. Pl do two-way ANOVA considering replication also as factor. Pl also describe the regression. You may attempt stepwise regression. You may refer Large Scale Spatial Assessment, Modelling and Identification of Drivers of Soil Respiration.
Response: Thanks for your comments which are really helpful for our comprehensive consideration. The single factor affecting the variation of NEE, ER and GEP was analyzed by linear regression. We have used stepwise regression analysis to analyze the common factors affecting the variation of NEE, ER and GEP. Please see line 262-272.
- 9. Please mention how many measurements you have observed for each variable for each grasslands.
Response: Thanks for your comments. In this study, NEE, ER and GEP were intensively monitored for 21 times during the whole growth period of plants. NEE, ER and GEP were not measured from wintering to regreening because there were no plants on the ground.
- 10. Please provide the details of calibration for each used instruments.
Response: Thanks for your comments. As suggested, we have supplemented the details of calibration for each used instruments. The details of calibration for each used instruments has been revised as follows:
The portable photosynthetic system: Li-6800, Li-cor Inc., Lincoln, NE, USA.
The pHS-3G digital pH meter : from Shanghai LeiMag Instrument Factory, Shanghai, China.
The elemental analyzer: Elementar , Vario EL cube ,Germany.
The continuous flow analyzer: SEAL , AA3 , Germany.
- 11. Pl elaborate the quality of grasses i.e. growth and survivality etc.
Response: Thanks for your comments. In this study, During the full-bloom stage of Medicago sativa and Vicia villosa, the filling stage of Secale cereale and Triticum aestivum we measured the aboveground and underground biomass of plants and the total carbon content and total nitrogen content of plants. The specific data are as follows :
|
Aboveground biomass (g/m2) |
Aboveground biomass (g/m2) |
Total carbon content (g/kg) |
Total nitrogen content (g/kg) |
Medicago sativa |
1245.97±128.21a |
361.41±78.09a |
394.90±10.08b |
25.64±1.76a |
Triticum aestivum |
963.54±101.47b |
279.12±59.04ab |
408.86±3.26a |
11.34±1.43b |
Secale cereale |
1151.32±126.39ab |
402.94±144.99a |
417.95±5.57a |
10.57±1.87b |
Vicia villosa |
732.53±94.57c |
173.33±66.25b |
384.88±11.79b |
27.73±6.21a |
- 12. The result is presented well however I would like to see the causes of differences that should not be based simply the difference of the grasses rather other attributal causes also, if feasible.
Response: Thanks for your comment. We also measured the soil physicochemical properties of each grassland while measuring NEE, ER and GEP. Pearson correlation analysis and linear regression were applied to examine the relationships between CO2 flux (ER, NEE, and GEP) and soil properties. The result showed that soil bulk density was significantly negatively correlated with NEE and GEP. The soil bulk density of Medicago sativa grassland is the lowest. Because alfalfa develops a taproot, more roots, and longer roots, meaning that it can play a very good role in thinning soil during the development and extension of roots. Medicago sativa grassland had the smallest soil bulk density and the largest soil porosity, promoting the absorption of soil nutrients by plants and increasing the aboveground biomass, microbial activity, thus increasing NEE, and GEP. In this study, the main factor affecting ER is MBC. MBC can explain 60.1 % of ER changes. Soil microbial carbon content can reflect the quantity and activity of soil microorganisms, which directly affect the activity of soil microorganisms and their roots, and then affect soil respiration. When soil microbial carbon content increases, microbial activity and quantity increase, leading to the decomposition of organic matter by microorganisms. In this process, microorganisms consume organic matter through respiration. And release gases such as carbon dioxide, which leads to ER enhancement. The root system of alfalfa is developed, and the root system secretes a variety of substances during the growth process. These root exudates have a significant effect on the soil environment. The root exudates are rich in organic substances. These organic substances gradually accumulate in the soil, thereby increasing the content of soil organic carbon and microbial carbon, and providing sufficient carbon sources for the growth and reproduction of microorganisms, thus promotes the respiration of the aboveground and underground parts.
- 13. Please check title of each Table and figure. Figure 6 is scatter plot with regression equation. If you at all interested for evaluation the drivers, pl use syep wise regress. It would be good for better result.
Response: Thank you, the comments are very important to improve our manuscript. We have modified the title of each Table and figure in the revised manuscript.
- Though the discussion is attempted to account the result however poorly describe. Pl improve based on study on similar aspects in similar region.
Response: Thanks for your comments which are really helpful for our comprehensive consideration. We have revised the discussion part in the revised manuscript and marked it with red font. Please see line 273-351.
- How the study is useful for other regions needs to be included.
Response: Thanks for your comments which are really helpful for our comprehensive consideration. In this study, we found that the carbon sequestration capacity of alfalfa and ryegrass is strong, and the aboveground biomass and underground biomass are also large, which is conducive to improving the productivity of grassland. Therefore, these two kinds of grassland can be planted in a large area.
- Please include limitation and future researches directions.
Response: Thank you, the comments are very important to improve our manuscript. The limitation of this study is that due to time constraints, only the carbon flux of cultivated grassland in a single area was measured and the closed box method was used for monitoring. In the future, it is necessary to expand the research scope, conduct long-term monitoring, obtain more data to study the law of carbon flux change in cultivated grassland, and adopt different instruments and methods to monitor carbon flux, and various methods complement each other, so as to obtain more accurate carbon flux data.
- Please make precise the conclusion.
Response: Thank you, the comments are very important to improve our manuscript. We have revised the conclusion part in the revised manuscript and marked it with red font. Please see line 352-369.
Thanks again for your attention and time. Look forward to hearing from you.
Yours sincerely,
Zhenchao Zhang
25 April, 2024
Email: [email protected]

Reviewer 2 Report
Comments and Suggestions for Authors
The study is carefully designed, implemented, and reported.
The study does not appear to discuss any organic solid matter flow out of the grasslands. Such flow possibly is not included in the experiment. However, in real-life applications, there is harvesting of the grass crops, which are then utilized as animal forage, or for technical purposes. If the crops are foraged by animals, the stored carbon is released back to the atmosphere. In a stationary state, a grassland ecosystem performs no net carbon sequestration. If there is accumulation of carbon on the soil, the system is not in a stationary state, and there may be carbon sequestration.
If there is outflow of stored carbon in terms of harvested grass crops, the carbon sequestration cannot be determined by gas exchange analysis only.
Author Response
Dear reviewer:
Thank you for your letter and for the reviewers’ comments concerning our manuscript entitled " Cultivated Grassland Types Differently Affected Carbon Sequestration Downstream of the Yellow River" (ID:agronomy-2964681). Those comments are all valuable and very helpful for revising and improving our paper, as well as the important guiding significance to our researches. We have studied comments carefully and have made correction which we hope meet with approval.
Responds to the reviewer’s comments:
The study does not appear to discuss any organic solid matter flow out of the grasslands. Such flow possibly is not included in the experiment. However, in real-life applications, there is harvesting of the grass crops, which are then utilized as animal forage, or for technical purposes. If the crops are foraged by animals, the stored carbon is released back to the atmosphere. In a stationary state, a grassland ecosystem performs no net carbon sequestration. If there is accumulation of carbon on the soil, the system is not in a stationary state, and there may be carbon sequestration.
If there is outflow of stored carbon in terms of harvested grass crops, the carbon sequestration cannot be determined by gas exchange analysis only.
Response: Thanks for your comments which are really helpful for our comprehensive consideration. It is true we have not consider the organic solid matter among different grasslands. The main purpose of this study was to study the variation characteristics of carbon flux during the growth period of different cultivated grassland. Net ecosystem CO2 exchange (NEE) studies in grassland ecosystems have helped to confirm the balance of the amount of carbon sinks in soil and the volume of carbon sources of CO2 in atmosphere. The amount of carbon sinks is determined by plants during photosynthetic activity processes (Gross ecosystem productivity, GPP) and the volume of carbon sources is determined by plants respiration and heterotrophic respirationduring their respiration processes (ecosystem respiration, ER). In the next study, we will analyze the carbon sequestration capacity of different cultivated grasslands through the combination of carbon flux and soil carbon storage, so as to provide a theoretical basis for the carbon pool management of cultivated grasslands downstream of the Yellow River. According to your suggestion, we have also changed the title of this paper to “Cultivated Grassland Types Differently Affected Carbon Flux Downstream of the Yellow River”. Thank you very much for your valuable advice.
Thanks again for your attention and time. Look forward to hearing from you.
Yours sincerely,
Zhenchao Zhang
25 April, 2024
Email: [email protected]

Round 2
Reviewer 1 Report
Comments and Suggestions for Authors
Good efforts
Reviewer 2 Report
Comments and Suggestions for Authors
Gongratulations.